# Elucidating the potential carcinogenic molecular mechanisms of parabens in head and neck squamous cell carcinoma through network toxicology and molecular docking

**Lei Zhao, Jianwang Yang, Tao Liu, Huan Cao, Miaomiao Yu, Baoshan Wang** ✩ *

Department of Otolaryngology-Head and Neck Surgery, The Second Hospital of Hebei Medical University, Shijiazhuang, Hebei, China

* wangbaoshan@hebmu.edu.cn

## Abstract

### Objective

This study aims to systematically investigate the molecular mechanisms through which parabens may contribute to head and neck squamous cell carcinoma (HNSCC) carcinogenesis using integrated network toxicology and molecular docking.

### Materials and methods

Six commonly used parabens (ethyl-, propyl-, methyl-, heptyl-, butyl-, and benzylparaben) were selected for toxicity prediction via ProTox 3.0 and ADMETlab2.0. Their potential targets were retrieved from Swiss Target Prediction, ChEMBL, and the Similarity Ensemble Approach (SEA). HNSCC-related targets were collected from GeneCards, OMIM, and CTD, while differentially expressed genes (DEGs) were identified using TCGA-HNSC data. Functional enrichment, protein-protein interaction (PPI) network construction, hub gene identification, molecular docking, and immune infiltration analyses were performed using DAVID, STRING, Cytoscape, CB-DOCK2, and TIMER2.0.

### Results

We identified 80 common targets through which parabens may exert toxic effects in HNSCC. Among these, 12 hub genes (CCNB1, CDK1, CCNA2, CDK2, CDK4, TYMS, AURKA, CCNA1, CHEK1, CCNB2, PLK1, CDC25A) were significantly overexpressed in HNSCC tissues and were primarily enriched in cell cycle regulation, p53 signaling, and viral carcinogenesis pathways. CCNA1 was further validated as an independent prognostic factor associated with poor survival. Molecular docking revealed strong binding affinities between parabens and hub proteins. Immune infiltration analysis indicated a negative correlation between CCNA1 expression and CD8+T cell and B cell infiltration.

**Data availability statement:** All relevant data are within the manuscript and its Supporting information files.

**Funding:** This research was funded by the Clinical Research Center Foundation of Hebei Provincial Department of Science and Technology, grant number 20577716D, and the Natural Science Foundation of Hebei Province, grant number H2022206376.

**Competing interests:** The authors have declared that no competing interests exist.

## Conclusions

Parabens may promote HNSCC progression by disrupting cell cycle regulation and immune responses via direct interactions with key hub genes. These findings provide a novel mechanistic basis for the carcinogenic potential of parabens in HNSCC and underscore the need for further experimental and epidemiological validation.

## Introduction

Parabens are among the most widely used synthetic preservatives globally. Their broad-spectrum antimicrobial efficacy and low cost have led to extensive application in cosmetics, pharmaceuticals, food products, and personal care items (PCPs) [1–5]. Human exposure to parabens is continuous and nearly universal, occurring primarily through dermal absorption and dietary intake [5–7]. Growing evidence suggests potential adverse effects, including associations with cancers, diabetes, thyroid diseases, and obesity, raising public health concerns [3,6–11].

Parabens are well-established endocrine-disrupting chemicals (EDCs) with estrogen-mimicking properties. They bind to and activate estrogen receptors (ERs), potentially disrupting hormonal homeostasis, reproductive functions, and developmental processes [1,12–15]. Numerous in vitro and in vivo studies have linked parabens to developmental toxicity, reproductive disorders, and allergic skin reactions [16–20]. Of particular relevance to carcinogenesis is the ability of longer-chain parabens, such as butylparaben, to exhibit stronger estrogenic activity and promote proliferation in hormone-dependent cancer cells [12,21–24]. Although parabens are not classified as direct mutagens, their role as tumor promoters is under active investigation. Research has primarily focused on hormone-responsive cancers such as breast cancer, where intact parabens have been detected in tumor tissues and shown to drive proliferation and migration via ER-mediated pathways and oncogenic signaling cascades like MAPK/ERK and PI3K/Akt [12,21,25–27]. In contrast, their impact on non-hormonally driven cancers, including head and neck cancers, remains poorly understood [28,29].

Parabens are consistently detected in environmental matrices worldwide due to extensive consumption and incomplete removal during wastewater treatment [30–33]. They have been found in human urine, breast tissue, and adipose tissue, confirming widespread exposure [7,30,31]. Biomonitoring studies detect paraben metabolites in over 90% of urine samples from general populations [34,35], highlighting the need for comprehensive toxicological risk assessment.

Head and neck squamous cell carcinoma (HNSCC) ranks as the sixth most common cancer globally, with a five-year survival rate that has remained stagnant for decades [36,37]. GLOBOCAN estimates indicate a significant global impact from HNSCC in 2025, with projected figures of 946,456 new cases and a mortality count of 482,001 [38]. Established risk factors include tobacco use, alcohol consumption, and high-risk human papillomavirus (HPV) infection [36,37]. However, many cases, especially among young adults without traditional risk factors, suggest a role for other

environmental or occupational exposures [36,39]. Despite advances in treatment, the molecular pathogenesis of HNSCC remains complex and heterogeneous, frequently involving aberrant activation of EGFR, NF-κB, and PI3K/Akt pathways, as well as dysregulation of cell cycle and apoptosis mechanisms [40–46].

Recent evidence indicates that parabens can modulate chemotaxis, adhesion, and phagocytosis in macrophages [2], which play a dual regulatory role in HNSCC tumor immunology [47]. The potential contribution of parabens to HNSCC initiation or progression represents a compelling yet underexplored research area.

Thus, a critical gap exists between the well-documented widespread human exposure to parabens, their suggested tumor-promoting capabilities in other cancers, and the scarcity of research on their biological effects in HNSCC. It remains unknown whether parabens can influence hallmark traits of HNSCC, such as proliferation, invasion, metastasis, or therapy resistance, or what underlying molecular mechanisms might be involved. Addressing this gap is essential for a comprehensive understanding of HNSCC etiology and for informing public health policies on the safety of these common preservatives. Focusing on six commonly used parabens (ethyl-, propyl-, methyl-, heptyl-, butyl-, benzylparaben) [1,2], this study aims to bridge this gap by employing network toxicology integrated with molecular docking to systematically predict and elucidate the molecular mechanisms through which parabens may influence HNSCC development and progression.

## Materials and methods

### Toxicity prediction and analysis of common parabens

Canonical SMILES sequences of the six parabens were obtained from PubChem (https://pubchem.ncbi.nlm.nih.gov/) [48] and submitted to ProTox 3.0 (https://tox.charite.de/protox3/) [49] and ADMETlab2.0 (https://admetmesh.scbdd.com/service/evaluation/cal) [50] for toxicity prediction.

### Retrieval of paraben targets

Potential protein targets of the six parabens were identified using Swiss Target Prediction (http://www.swisstargetprediction.ch/) [51] (Homo sapiens, probability > 0), ChEMBL (https://www.ebi.ac.uk/chembl/) [52,53] (Homo sapiens), and the Similarity Ensemble Approach (SEA) (https://sea.bkslab.org/) [54] (Homo sapiens, probability > 0). SMILES sequences were used as input for Swiss Target Prediction and SEA, while PubChem names were used for ChEMBL queries. Targets were normalized using UniProt (https://www.uniprot.org/) [55,56].

### Retrieval of HNSCC-related targets

HNSCC-associated targets were collected from GeneCards (https://www.genecards.org/) [57], Online Mendelian Inheritance in Man (OMIM) (https://omim.org/) [58], and Comparative Toxicogenomics Database (CTD) (https://ctdbase.org/) [59] using the keywords: "Squamous Cell Carcinoma of Head and Neck", "HNSCC", "Head and Neck Squamous Cell Carcinoma", "Oral Cavity Squamous Cell Carcinoma", "Squamous Cell Carcinoma of the Larynx", "Laryngeal Squamous Cell Carcinoma", "Squamous Cell Carcinoma of the Nasal Cavity", "Oropharyngeal Squamous Cell Carcinoma", "Hypopharyngeal Squamous Cell Carcinoma", and "Tongue Squamous Cell Carcinoma".

### Acquisition of HNSCC-related targets from TCGA

Differential gene expression profiles and clinical phenotype data were downloaded from The Cancer Genome Atlas (TCGA) (https://www.cancer.gov/tcga) [60,61] via the Home for Researchers platform (https://www.home-for-researchers.com/). Differentially expressed genes (DEGs) were defined as those with adjusted p-value < 0.05 and |log2(fold change)| ≥ 1. Gene Ontology (GO) (http://geneontology.org/) [62,63] and Kyoto Encyclopedia of Targets and Genomes (KEGG) pathway (https://www.kegg.jp) [64,65] enrichment analyses were also completed with the same platform.

### Identification of common targets for parabens and HNSCC

Overlapping targets among paraben-related targets, HNSCC-related targets, and TCGA-derived HNSCC DEGs were identified using jvenn (https://jvenn.toulouse.inrae.fr/app/example.html) [66]. The resulting common targets were considered candidate targets for paraben-induced toxicity in HNSCC. Functional enrichment analyses (GO, KEGG, and REACTOME Pathway (https://reactome.org/) [67,68])were conducted using Database for Annotation, Visualization, and Integrated Discovery (DAVID) (https://david.ncifcrf.gov/) [69,70] with default parameters (OFFICIAL GENE SYMBOL, Homo sapiens). Results were visualized via circle enrichment plots generated with ChiPlot (https://www.chiplot.online/).

### Construction of protein-protein interaction network

A protein-protein interaction (PPI) network for the candidate targets of parabens in HNSCC was constructed using STRING (v12.0; https://cn.string-db.org/) [71] under the following settings: organisms "Homo sapiens", network type "full STRING network", medium confidence score > 0.4, and medium FDR stringency (5%).

### Identification of hub genes from the PPI network

The PPI network was imported into Cytoscape (v3.9.1) [72]. The MCODE plugin [73] was used to identify central clusters with default parameters (Degree Cutoff = 2, Haircut clustering, Node Score Cutoff = 0.2, K-Core = 2, and Max Depth = 100). Key targets from MCODE modules were compared with top-ranked targets from CytoHubba [74]. The top 20 genes from each of the 12 CytoHubba algorithms (Betweenness, BottleNeck, Closeness, ClusteringCoefficient, Degree, DMNC, EcCentricity, EPC, MCC, MNC, Radiality, and Stress) were selected [75]. An upset plot generated via ChiPlot identified the 20 most frequently occurring genes across all 12 algorithms. Final hub genes were defined as the intersection between MCODE-derived core targets and the top-ranked CytoHubba targets.

### Expression validation of hub genes in HNSCC

Protein expression levels of hub genes were evaluated using the Human Protein Atlas (HPA) (https://www.proteinatlas.org/) [76] and the University of ALabama at Birmingham CANcer data analysis Portal (UALCAN) (https://ualcan.path.uab.edu/) [77,78], following established methodologies [79,80]. Data analyzed by UALCAN were generated by the National Cancer Institute Clinical Proteomic Tumor Analysis Consortium (CPTAC) [81–83]. Gene expression profiles were obtained from the Gene Expression Omnibus (GEO) Datasets (https://www.ncbi.nlm.nih.gov/gds) [84,85] under the filters: expression profiling by array, tissue attribute, Homo sapiens, and paired samples only. Statistical differences were assessed and visualized using GraphPad Prism v9.0.0 (San Diego, California USA, www.graphpad.com). Additional validation was performed using Gene Expression Profiling Interactive Analysis (GEPIA3) (https://gepia3.bioinfoliu.com/) [86] with |log2(fold change) ≥ 1, q-value < 0.05, and TCGA paired peritumor samples as normal controls.

### Construction of a paraben exposure risk model for HNSCC

The association between hub gene expression and overall survival in HNSCC patients was evaluated using GEPIA3. Univariable and multivariate Cox models were applied using overall survival in months, hazard ratio (HR) calculation, median group cutoff, and HNSC Tumor datasets.

### Enrichment analysis of hub genes

GO and KEGG enrichment analyses for the hub genes were performed using DAVID with default parameters as above-mentioned and results were visualized with ChiPlot.

### Immune cells infiltration

The relationship between hub gene expression and tumor-infiltrating immune cell abundance was assessed using the TIMER2.0 (http://timer.cistrome.org/) [87–89]. Six computational algorithms (TIMER, XCell, MCP-COUNTER, CIBER-SORT, EPIC, and QUANTISEQ) were employed to estimate immune infiltration levels. Partial Spearman's correlation, adjusted for tumor purity, was used to evaluate correlations between gene expression and immune cell abundance. Correlations with $\rho > 0$ and $p < 0.05$ were considered significantly positive; those with $\rho < 0$ and $p < 0.05$ were deemed significantly negative; and results with $p \geq 0.05$ were considered non-significant.

### Molecular docking

Molecular docking was performed to evaluate binding affinities between core proteins and paraben. The three-dimensional structures of the target proteins were retrieved from the RCSB Protein Data Bank (PDB) (https://www.rcsb.org/) [90,91] for experimentally-determined structures and AlphaFold Protein Structure Database (https://alphafold.ebi.ac.uk/) [92,93] for computationally predict structures, while the paraben structures in SDF format were obtained from PubChem (https://pubchem.ncbi.nlm.nih.gov) [48,94]. Subsequently, both the receptor and ligand files were imported into CB-DOCK2 (https://cadd.labshare.cn/cb-dock2/php/index.php) [95,96] for automated blind docking simulations.

### Compound-target-pathway network

A Sankey diagram was constructed using ChiPlot to visualize the multi-layered relationships between the six parabens (source nodes), hub genes (middle nodes), and enriched KEGG pathways (target nodes). Linkage data defining the connections between these nodes were formatted into a source-target-weight matrix and uploaded to ChiPlot for visualization.

## Results

### Toxicity prediction and identification of paraben-related targets

Toxicity predictions for the six parabens were obtained from ProTox 3.0 and ADMETlab2.0 (retrieved 2025-7-28), revealing varying degrees of potential toxicity (S1 Table). A total of 582 unique human targets related to parabens were identified after merging and deduplicating results from Swiss Target Prediction (234 targets), ChEMBL (137 targets), and SEA (342 targets) (retrieved 2025-7-29). Heptylparaben and benzylparaben were not available in ChEMBL. A flowchart schematically illustrated the refined screening workflow (Fig 1).

### Screening of HNSCC-related targets

HNSCC-related targets were identified from GeneCards (447 protein-coding genes after removing non-coding RNAs from 852 predicted targets), OMIM (1,352 targets), CTD (6,242 targets, including 44 with curated associations to HNSCC and 6,198 with inferred associations and inference scores above the average value of 7.11 (retrieved 2025-8-3). After merging and deduplication, 7,320 HNSCC-related targets were obtained.

### Analysis of differential expressed genes in HNSCC from TCGA

Analysis of RNA-seq data from 504 HNSCC tumors and 44 matched normal samples from TCGA identified 2,250 DEGs (1,623 up-regulated, 627 down-regulated; adjusted $p < 0.05$, $|\log_2 FC| \geq 1$) (retrieved 2025-8-4). The distribution of DEGs was shown in a volcano plot (S1A Fig), and the top 50 up- and down-regulated genes were presented in a heatmap (S1B Fig). In order to intuitively study the shared functions or characteristics of DEGs, GO and KEGG analyses were presented in a circle enrichment plot (S1C Fig). Furtherly, separate enrichment analyses for up- and down-regulated DEGs were presented in S1D–S1G Fig. Notably, the highest number of up-regulated DEGs (80 genes) were enriched in human papillomavirus infection pathway (hsa05165), consistent with known etiology of HPV-positive HNSCC (S1E Fig).

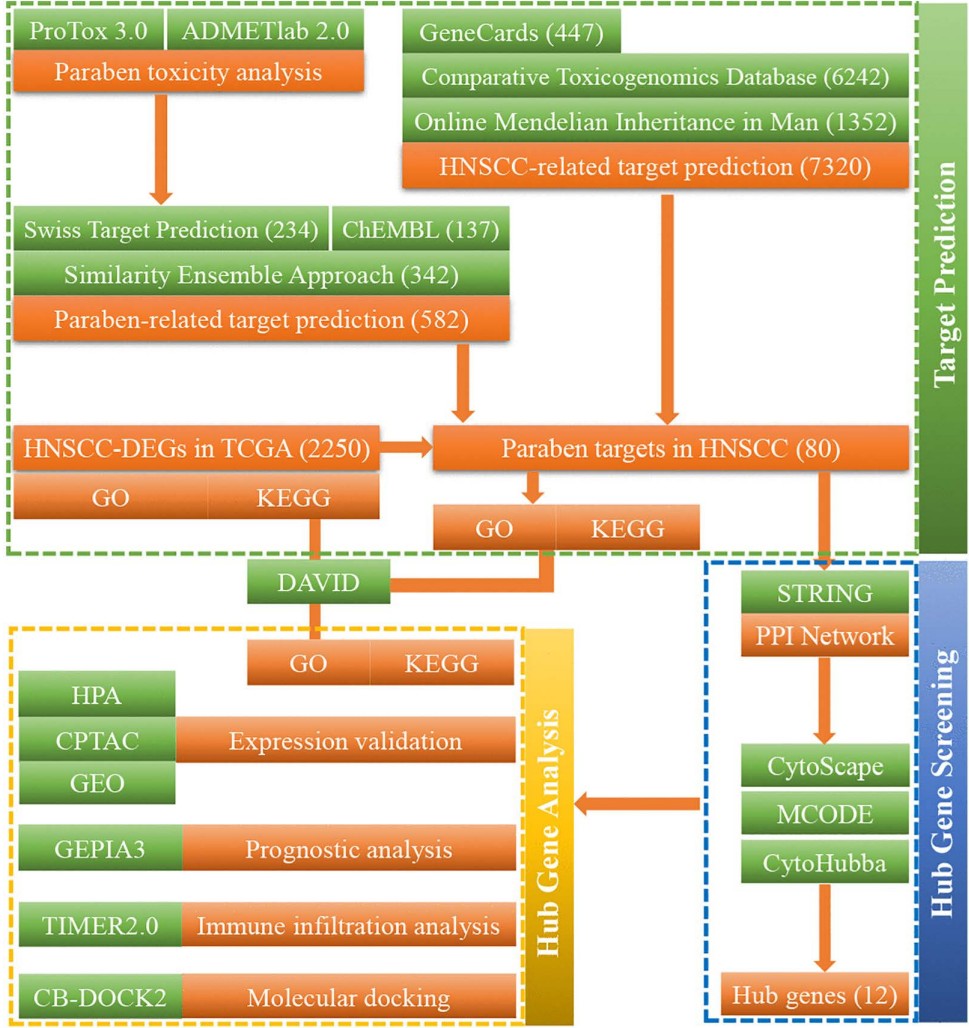

**Fig 1.  Schematic flowchart of the integrated network toxicology and molecular docking study on paraben toxicity in HNSCC.**

## Identification of parabens-related targets in HNSCC and construction of PPI network

Intersection analysis identified 80 candidate targets through which parabens may exert toxic effects in HNSCC (Fig 2 and S2 Table). Then, enrichment analysis indicated that these targets were associated with 298 GO terms, comprising 181 Biological Process (BP), 56 Cellular Component (CC), 61 Molecular Function (MF). Analysis also shown that these targets enriched in 42 KEGG terms, and 120 REACTOME pathways. The top 10 terms ranked by p-value for each category were visualized in circle enrichment plots (Fig 3 and S3 Table) (retrieved 2025-8-7). The PPI network constructed from the 80 candidate targets contained 569 edges among 80 nodes, with an average node degree of 14.2 and an average local clustering coefficient of 0.634), indicating significant enrichment over random expectation (expected edges = 208, $p < 1.0 \times 10^{-16}$) (S2 Fig) (retrieved 2025-8-7).

## Identification of hub genes for parabens targeting HNSCC

MCODE analysis identified a core subnetwork with the highest score (19.905), comprising 22 nodes and 418 edges (Fig 4A). CytoHubba identified the top 20 genes from each of 12 algorithms. Further, an upset plot analysis revealed the

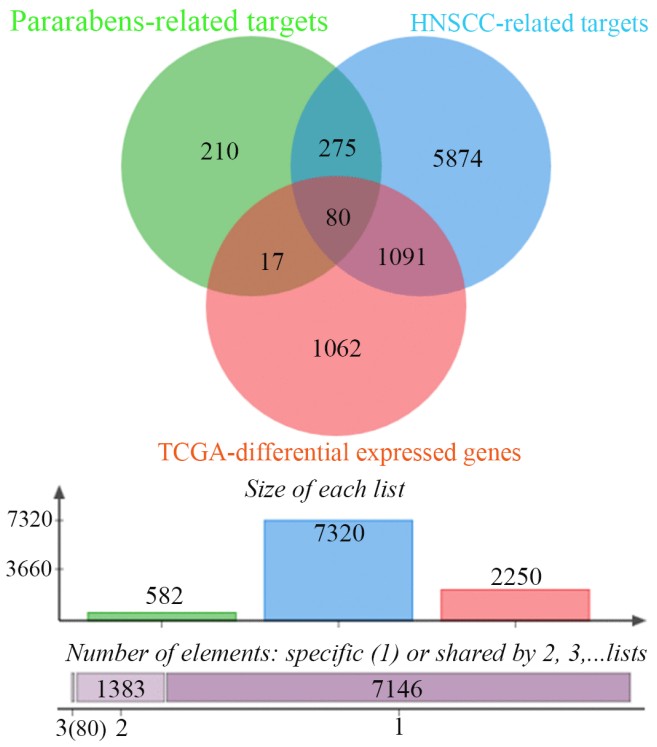

Parabens-related targets    HNSCC-related targets

TCGA-differential expressed genes

*Size of each list*

*Number of elements: specific (1) or shared by 2, 3,...lists*

**Fig 2. Identification of common candidate targets for paraben toxicity in HNSCC.** A three-way Venn diagram illustrates the overlap among three distinct datasets: 582 potential paraben-related targets from Swiss Target Prediction, ChEMBL, and the Similarity Ensemble Approach (SEA); 7,320 known HNSCC-associated genes from GeneCards, Online Mendelian Inheritance in Man (OMIM), and Comparative Toxicogenomics Database (CTD); and 2,250 differentially expressed genes (DEGs) between HNSCC tumors and normal tissues from The Cancer Genome Atlas (TCGA) cohort. The intersection of all three sets reveals 80 common candidate genes, which are hypothesized to be critically involved in the potential toxicological mechanisms of parabens on the development and progression of HNSCC.

top 20 most frequently occurring genes across all algorithms (Fig 4B). Among these genes, 12 genes (CCNB1, CDK1, CCNA2, CDK2, CDK4, TYMS, AURKA, CCNA1, CHEK1, CCNB2, PLK1, CDC25A) were located within the above-mentioned core network and were designated as hub genes (Fig 4).

## Expression validation of hub genes in HNSCC

The protein expression levels and subcellular location of hub genes were assessed between oral mucosa and HNSCC tissues using immunohistochemistry in HPA database (retrieved 2025-8-8). Of the 12 hub proteins, 10 were retrieved in the HPA database, while CHEK1 and CDC25A were not included in it. Among the 10 proteins, the expression levels of CCNB1, CDK4, TYMS, and CCNA1 exhibited higher expression in HNSCC tissues compared to oral mucosa (S3 Fig). Besides, the hub proteins primarily localized in the cytoplasm and/or nucleus (S3 Fig). The protein expression levels of hub genes were further assessed by UALCAN (retrieved 2025-8-8). Results indicated that among the 12 proteins, nine proteins (CCNB1, CDK1, CCNA2, CDK2, TYMS, AURKA, CHEK1, CCNB2, PLK1) were over-expressed in HNSCC, while CDK4 showed no significant difference (S4 Fig). CCNA1 and CDC25A were not included in CPTAC data. Through search details: HNSCC [All Fields] AND "Homo sapiens"[porgn] AND ("Expression profiling by array"[Filter] AND "attribute name tissue"[Filter]) (retrieved 2025-8-9), the present study obtained 30 datasets. After reviewing the titles and abstracts of the 30 datasets, and focusing on paired-samples, GSE83519 (GPL4133, consist of 22 paired HNSCC tumor and normal tissues), GSE58911 (GPL6244, consist of 15 paired tumor and normal tissues) [97], and GSE160042 (GPL18180, consist

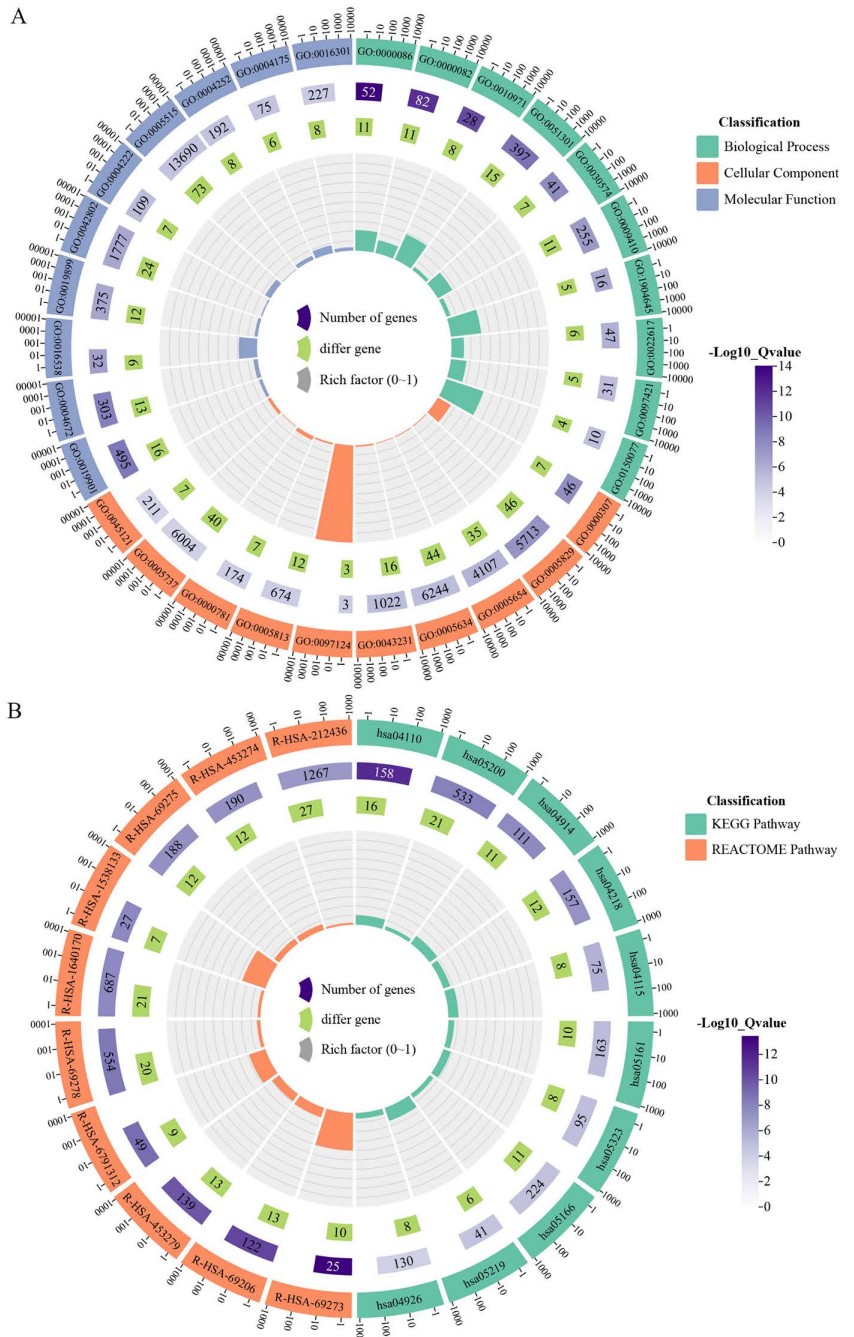

**Fig 3. Functional enrichment analysis of differentially expressed genes (DEGs) in the TCGA-HNSCC cohort.** A circular enrichment plot illustrates the top 10 most significantly enriched terms (by p-value) from Gene Ontology (GO), Kyoto Encyclopedia of Genes and Genomes (KEGG), and REACTOME pathway analyses for the HNSCC-DEGs. In these plots: the outermost circle lists the significantly enriched term IDs, with different colors representing different categories. The second circle represents the total number of background genes that enriched in each item. The third circle represents the genes in DEGs list that enriched in the target term. The innermost circle represents the rich factor, which is the ratio of enriched genes to total background genes, reflecting the degree of enrichment of the target pathway.

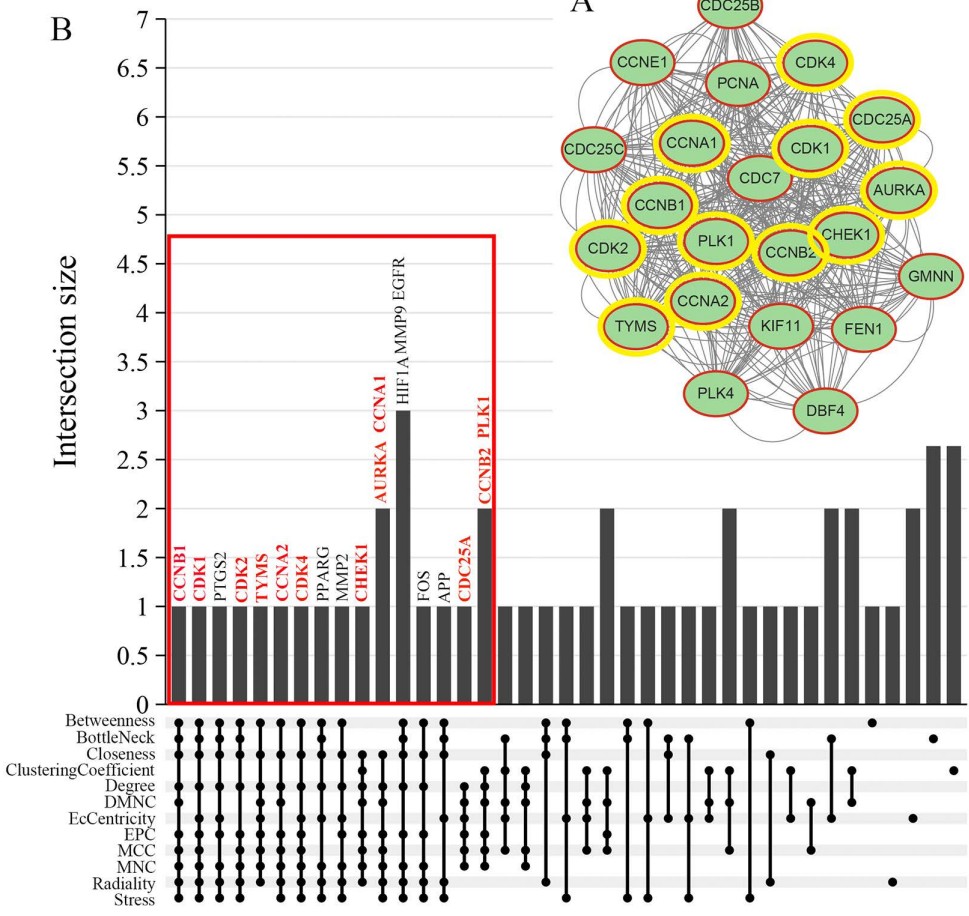

**Fig 4. Identification of hub genes via MCODE and CytoHubba analysis. (A)** A core protein-protein interaction (PPI) subnetwork was extracted using the MCODE algorithm, with the highest MCODE score (19.905), containing 22 nodes and 418 edges, indicating a highly interconnected protein cluster. **(B)** An upset plot shows the overlap of the top 20 genes ranked by each of the 12 CyroHubba algorithms. Horizontal bars represent the frequency of each gene across all methods. From this analysis, 12 hub genes (CCNB1, CDK1, CCNA2, CDK2, CDK4, TYMS, AURKA, CCNA1, CHEK1, CCNB2, PLK1, CDC25A) were not only highly ranked but were also all members of the core MCODE subnetwork, confirming their status as central hub genes for further investigation.

of 10 paired tumor and normal tissues) [98] were used for validating the expression levels of the hub genes. The expression matrix of each GEO Datasets above-mentioned were downloaded, and expression values of each hub gene were extracted. Then, the difference of hub genes in HNSCC were identified with GraphPad Prism. The vast majority of hub genes were up-regulated in HNSCC (Fig 5). Additionally, GEPIA3 confirmed the overexpression of all 12 hub genes in HNSCC cancerous tissues compared to normal tissues (S5 Fig).

**Prognostic significance of hub genes for HNSCC**

Univariable survival analysis using GEPIA3 (retrieved 2025-8-16) showed that only CCNA1 was significantly associated with patient survival (hazard ratio [HR] = 1.60, 95% confidence interval [CI]: 1.22–2.10, p = 0.001) (S6 Fig). The remaining 11 hub genes did not exhibit significant prognostic value (p > 0.05). Multivariable Cox regression model incorporating all 12 hub genes confirmed CCNA1 as an independent predictor of poor prognosis (adjusted HR = 1.097, 95% CI: 1.021–1.179,

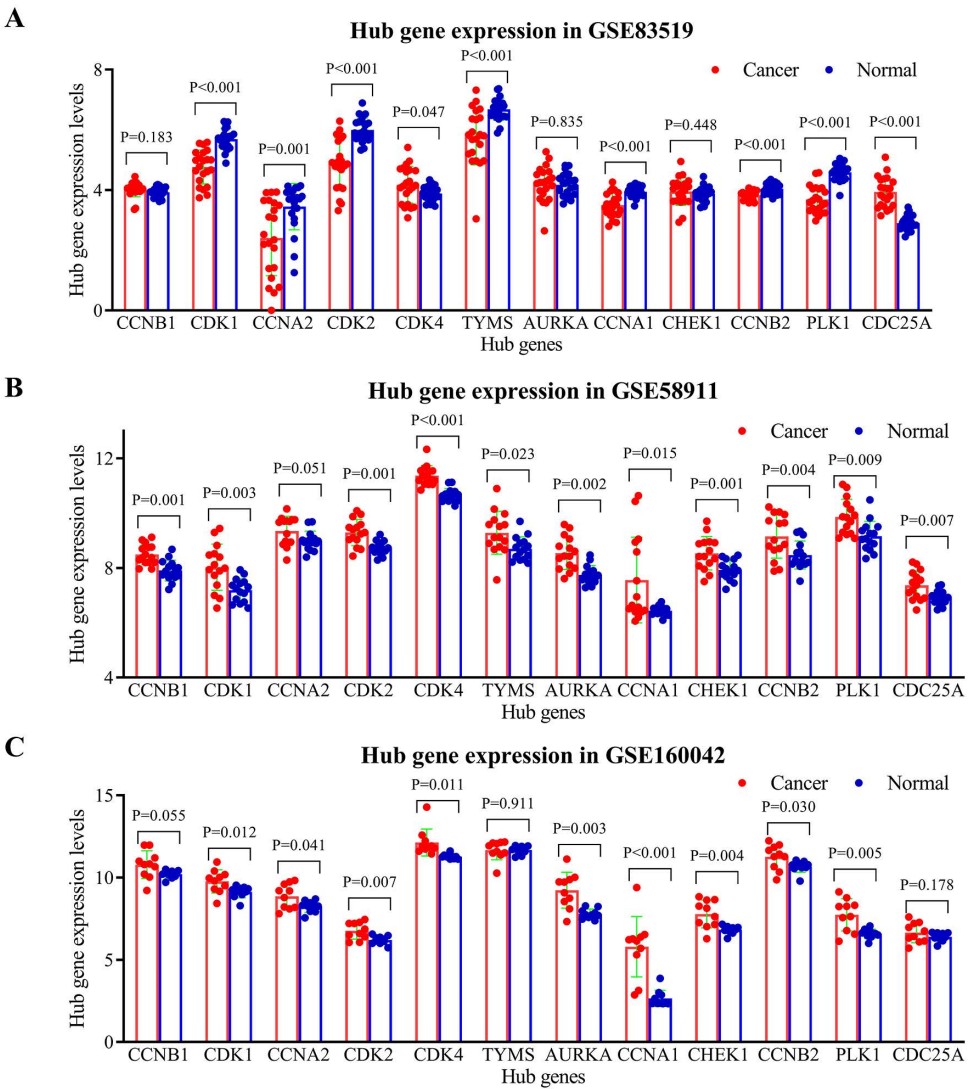

**Fig 5. Validation of hub gene expression in HNSCC using independent GEO datasets.** The mRNA expression levels of the identified hub genes were analyzed in three independent HNSCC cohorts from the Gene Expression Omnibus (GEO) repository. **(A)** GSE83519 (22 paired tumor and adjacent normal tissues), Note: A constant value (C = 3.911890567) was added to the raw expression data prior to $\log_2$ transformation to eliminate negative values. This constant value was calculated as |min (negative value)| + 0.001 to enable transformation of negative values; **(B)** GSE58911 (15 paired samples); **(C)** GSE160042 (10 paired samples). Differential expression between tumor and matched adjacent normal tissues is shown for each dataset. Statistical significance was determined by paired t-test (p < 0.05).

p = 0.012) (Fig 6). Notably, CCNA1 was the sole gene consistently significant in both analytical approaches, underscoring its strong and independent association with survival outcomes in this cohort.

## Enrichment analysis of the hub genes

Using DAVID database, functional enrichment analysis was performed on the 12 hub genes and yielded 67 significant GO terms (35 BP, 21 CC, 11 MF), 13 significant KEGG pathways, and 103 significant REACTOME pathways (S4 Table) (retrieved 2025-8-16). The top 10 enriched terms from each category were visualized in circular enrichment plots (Fig 7).

 

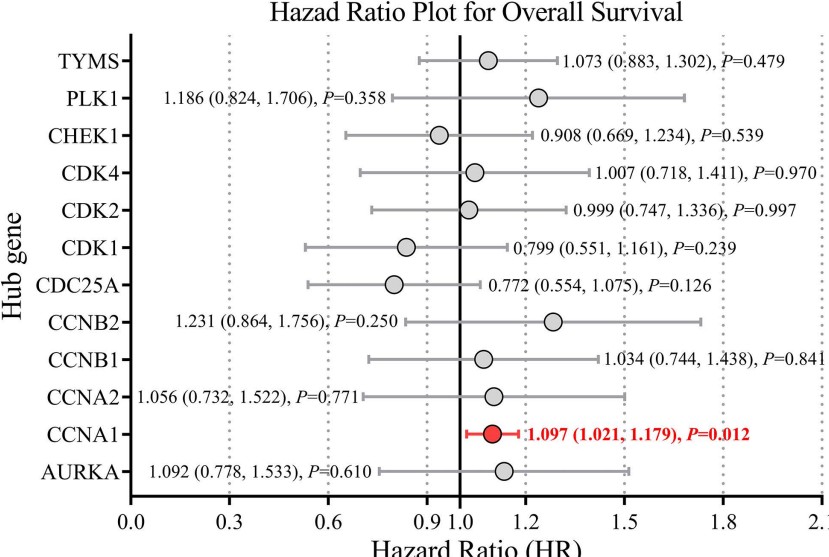

**Fig 6. Multivariable Cox regression analysis of the 12 hub genes for prognostic prediction in HNSCC.** A forest plot presents results of multi-variable Cox proportional hazards regression model incorporating all 12 candidate hub genes. Each gene is shown with its hazard ratio (HR) and 95% confidence interval (CI). CCNA1 was identified as a significant independent predictor of poor overall survival in months (adjusted HR = 1.097, 95% CI: 1.021–1.179, p = 0.012). No other hub genes showed significant independent prognostic value.

These findings indicated that the 12 hub genes were implicated in a range of coherent and biologically relevant processes, supporting their potential functional synergy. For example, among the 12 hub genes that were over-expressed in HNSCC, 10 were enriched in cell cycle pathway (hsa04110), six in p53 signaling pathway (hsa04115), and six in viral carcinogenesis pathway (hsa05203), all of which were related to HNSCC.

### Analysis of hub gene expression and immune cell infiltration

Analysis using TIMER2.0 (retrieved 2025-8-20) revealed correlations between the expression of the 12 hub genes and infiltration levels of six immune cell types (CD8 + T cells, CD4 + T cells, B cells, neutrophils macrophages, dendritic cells), which using partial Spearman's correlation to control for potential confounding factors such as tumor purity and clinical covariates. The results demonstrated that all 12 hub genes exhibited varying degrees of positive or negative correlations with different immune cell populations (S7 Fig). Notably, multiple algorithms consistently revealed that CCNA1, which is significantly associated with overall survival of HNSCC patients, showed a negative correlation with the infiltration levels of CD8 + T cells and B cells (Fig 8).

### Molecular docking between hub proteins and parabens

Molecular docking simulations using CB-DOCK2 demonstrated strong binding affinities between six paraben derivatives and the 12 hub proteins (Fig 9). Our analysis revealed that each paraben analogue engaged with multiple protein targets through specific hydrogen bonding and notable electrostatic interactions. Furthermore, the compounds consistently occupied hydrophobic regions within the target binding sites, suggesting stable complex formation. Notably, several interactions exhibited strong binding affinities, with energy values ranging between −8.1 and −9.2 kcal/mol, reflecting favorable and potentially physiologically relevant binding.

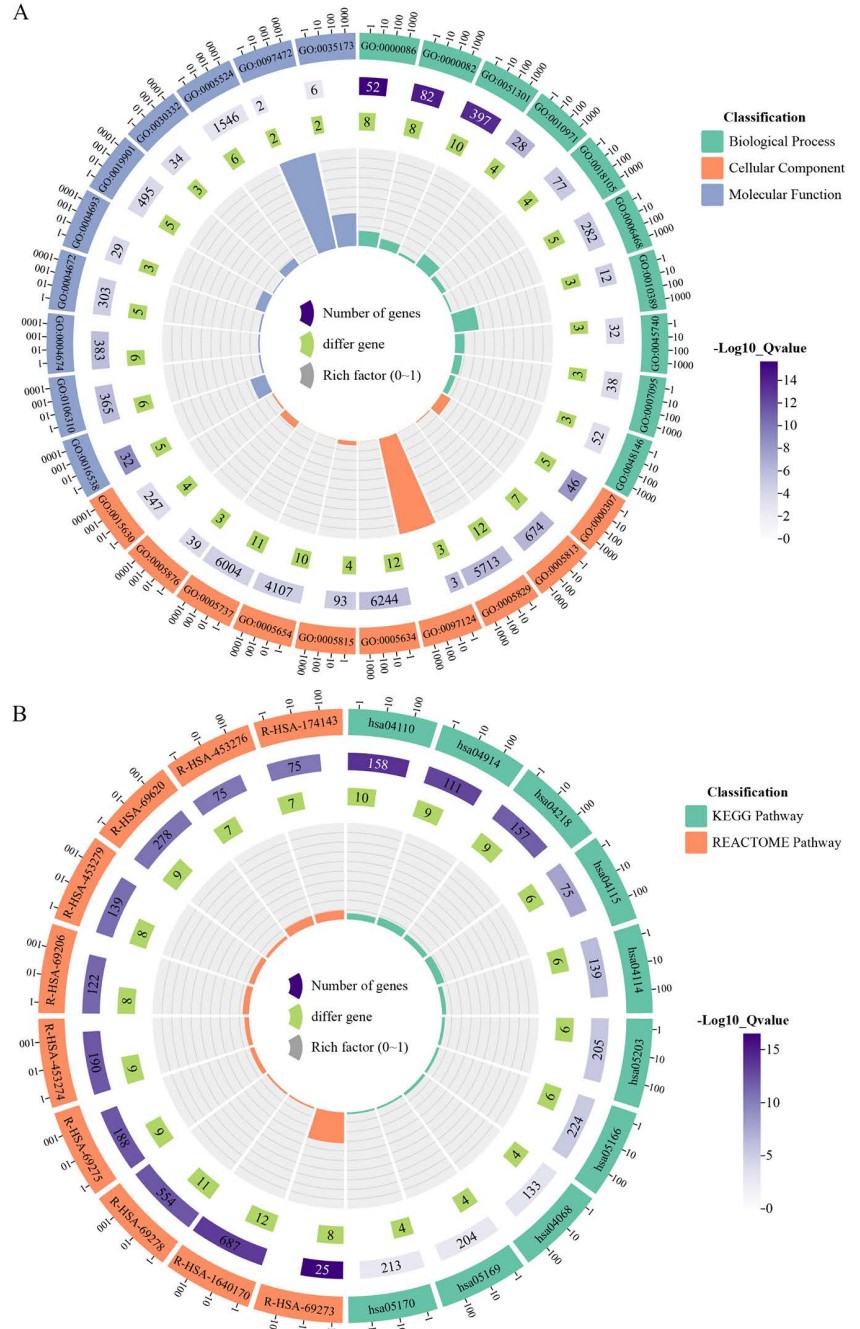

**Fig 7. Functional enrichment analysis of the 12 hub genes.** Circular enrichment plots visualize the top 10 most significantly enriched terms from **(A)** Gene Ontology (GO) and **(B)** Kyoto Encyclopedia of Genes and Genomes (KEGG) and REACTOME pathway analyses for the 12 hub genes.

## Construction of parabens-hub genes-pathways network

The Sankey diagram (Fig 10) illustrated the mechanistic flow from parabens exposure (left) through hub genes (center) to downstream functional pathways (right). The visualization confirmed the central role of the 12 hub genes as a convergent signaling layer, integrating inputs from multiple parabens and influencing critical biological pathways. For example,

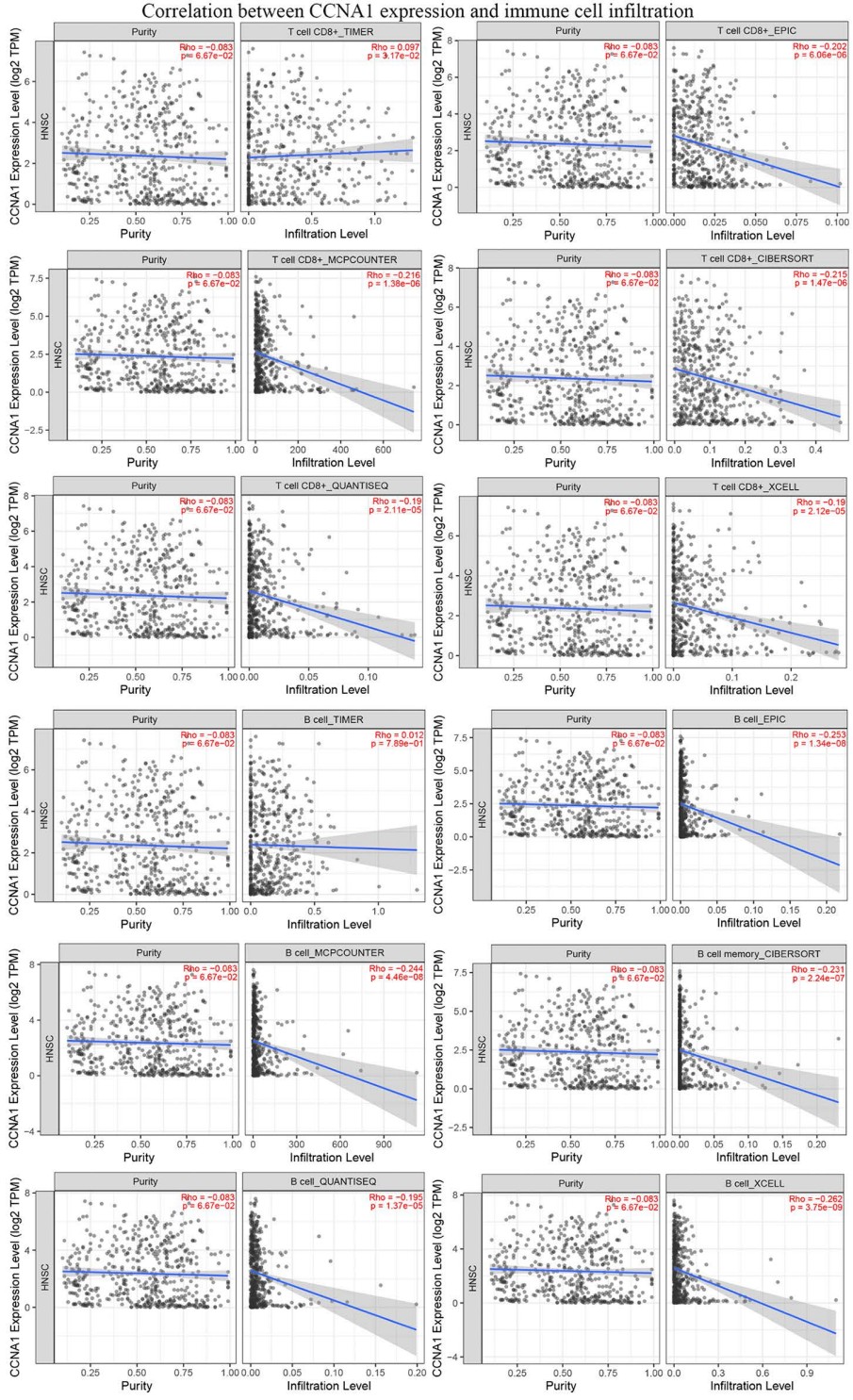

Correlation between CCNA1 expression and immune cell infiltration

**Fig 8. Association between CCNA1 expression and immune cell infiltration levels in TCGA-HNSC data.** Correlation between CCNA1 expression (log2(TPM + 1)) and infiltration levels of CD8 + T cell and B cell was analyzed across TCGA-HNSC tumors (n = 522) using TIMER2.0. Infiltration levels were estimated with six algorithms (EPIC, MCP-COUNTER, TIMER, CIBERSORT, QUANTISEQ, and XCELL) and adjusted for tumor purity. The association was evaluated using partial Spearman's correlation analysis. Spearman's ρ > 0 and p < 0.05 indicates significant positive correlation; ρ < 0 and p < 0.05 indicates significant negative correlation; p > 0.05 indicates no significance. TPM: transcripts per million. The analysis was performed by TIMER2.0 based on TCGA-HNSC data and detailed analysis can be further obtained from TIMER2.0.

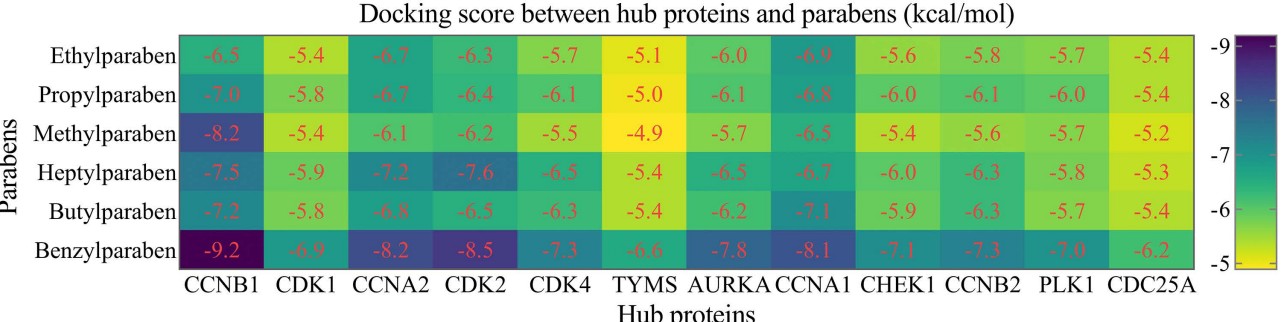

**Fig 9. Molecular docking analysis of paraben derivatives binding to hub proteins.** A structure-based blind protein-ligand docking strategy was performed using CB-DOCK2 to predict binding affinities between six paraben derivatives and 12 hub proteins. Results are presented as a heatmap, with the binding energy scores (Vina score, in kcal/mol) for each compound-protein pair indicated numerically. More negative scores represent stronger predicted binding affinity.

CCNA1 simultaneously mediated the exotic effects of benzyl-, butyl-, ethyl-, and methylparaben on pathways, including cell cycle, progesterone-mediated oocyte maturation, cellular senescence, viral carcinogenesis, human T-cell leukemia virus 1 infection, and Epstein-Barr virus infection. The Sankey diagram provides a powerful synthetic overview of the

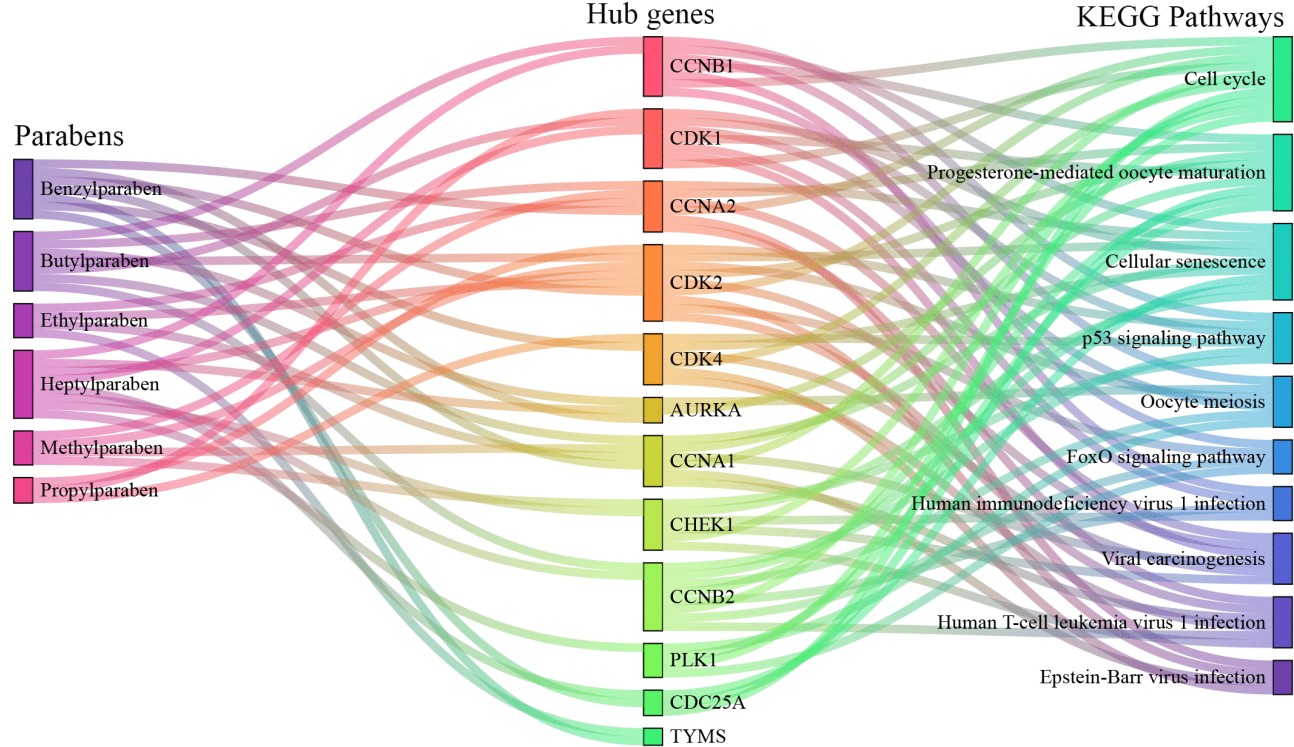

**Fig 10. Sankey diagram of paraben-hub gene-pathway interactions.** The diagram summarizes the proposed mechanistic links between paraben exposure (left), hub genes (middle), and enriched biological pathways (right), illustrating how parabens may influence HNSCC pathogenesis through core regulatory genes.

analysis, positing a coherent framework in which diverse paraben compounds modulated a core set of hub genes to perturb specific biological pathways.

## Discussion

This study systematically elucidates the potential molecular mechanisms through which parabens may exert toxic effects in HNSCC using integrated network toxicology and molecular docking approaches. Our findings reveal that parabens target key regulatory genes involved in cell cycle progression, DNA damage response, and viral carcinogenesis pathways, suggesting potential roles of parabens in HNSCC pathogenesis.

Through scientific and rigorous screening methods, we identified 12 hub genes (CCNB1, CDK1, CCNA2, CDK2, CDK4, TYMS, AURKA, CCNA1, CHEK1, CCNB2, PLK1, CDC25A) that are central to the PPI network of paraben-related targets in HNSCC and are significantly overexpressed in HNSCC tumor tissues compared to normal controls. Notably, CCNA1 emerged as an independent prognostic factor associated with poor survival in HNSCC patients, underscoring its clinical relevance. Enrichment analyses further indicated that these hub genes are functionally clustered in critical oncogenic pathways such as cell cycle regulation, p53 signaling, and viral carcinogenesis, which are frequently dysregulated in HNSCC [99–104]. Previous studies have also demonstrated that parabens can exhibit carcinogenic effects through multiple mechanisms [1,12,21,23,25].

Molecular docking results demonstrated strong binding affinities between parabens and these hub proteins, with favorable energy values indicating stable interactions. This suggests that parabens may directly modulate the activity of cell cycle regulators and DNA damage sensors, potentially leading to dysregulated proliferation and genomic instability, which are hallmarks of cancer. Particularly, the convergence of multiple parabens on CCNA1 linked exposure to altered cell cycle control and poor patient outcomes. Additionally, our immune infiltration analysis indicated that CCNA1 expression negatively correlated with CD8＋T cell and B cell infiltration, implying a potential immunosuppressive role of parabens. This aligns with recent evidence that tumor cell-intrinsic factors can shape the immune microenvironment in HNSCC, thereby influencing disease progression and therapy response [105–107]. While the estrogenic activity of parabens has been extensively studied in hormone-dependent cancers [12,21–24], their effects in non-hormonal cancers like HNSCC remained poorly understood. Our study provides a novel framework suggesting that parabens may promote carcinogenesis through non-canonical, ER-independent mechanisms, primarily by disrupting cell cycle checkpoints and enhancing chromosomal instability.

Noticeably, a key strength of this study lies in its integrative and systematic approach, combining multiple bioinformatics platforms, toxicological prediction tools, and molecular docking simulations to propose a novel mechanism of paraben-induced carcinogenesis in HNSCC. The use of publicly available multi-omics data from TCGA, GEO, CPTAC, and HPA enhances the reproducibility and reliability of the findings. Furthermore, the convergence of results across different analytical methods, including PPI network analysis, survival modeling, immune correlation assessment, and molecular docking, strengthens the validity of the identified hub genes and pathways. However, several limitations must be acknowledged. First, HNSCC is a heterogeneous group of diseases encompassing distinct anatomical subsites (such as pharynx, larynx, oral cavity) with different etiologies and molecular profiles. Our analysis was conducted using aggregated HNSCC data from public databases, which limits the ability to perform precise subsite-specific stratification without compromising statistical power. Therefore, our findings represent common pathways across HNSCC subtypes but may not fully capture site-specific mechanisms. Future studies should utilize TCGA subsite-specific cohorts (such as laryngeal cancer data) to validate and refine these findings. Second, the predictions made herein are primarily computational and require experimental validation using in vitro and in vivo HNSCC models to confirm causal relationships. Third, the study relies on existing databases that may contain inherent biases in sample selection or annotation. Additionally, the exact physiological concentration of parabens required to elicit these effects in human tissues remains unclear, and potential confounding factors such as exposure to other environmental carcinogens were not accounted for. Future work should

include dose-response experiments and prospective epidemiological studies to better quantify cancer risk associated with paraben exposure. Despite these limitations, this study provides a valuable theoretical framework and a prioritized list of candidate targets for subsequent investigations into the environmental toxicology of head and neck cancers.

Overall, this work provides a foundational hypothesis that parabens may contribute to HNSCC tumorigenesis by targeting core cell cycle genes and modulating immune responses. It highlights the need for further mechanistic studies and epidemiological investigations to clarify the role of paraben exposure in HNSCC etiology. These insights may eventually inform regulatory policies regarding the safe use of parabens in consumer products.

## Conclusions

In conclusion, this study provides the first comprehensive computational evidence that parabens may promote the pathogenesis of HNSCC through direct interactions with core cell cycle regulators and key signaling pathways. By integrating network toxicology and molecular docking, we identified 12 hub genes, which mediated paraben-induced toxicity in HNSCC and involved immune suppression. These findings shift the paradigm of paraben toxicity from a primarily estrogen receptor-driven mechanism to a broader oncogenic model involving cell cycle disruption and tumor microenvironment modulation. However, these insights remain predictive and necessitate further validation through in vitro and in vivo models, as well as epidemiological studies assessing real-world exposure levels. Ultimately, this work underscores the potential public health relevance of paraben exposure in HNSCC and supports the need for more stringent regulatory evaluation of these ubiquitous environmental chemicals.

## Supporting information

**S1 Fig. Transcriptomic analysis of head and neck squamous cell carcinoma (HNSC) from TCGA.** (A) Volcano plot of differentially expressed genes (DEGs) between HNSCC tumors and normal tissues. Red points indicate significant DEGs (adjusted p < 0.05 and |log2 fold change| > 1); gray points indicate non-significant genes. (B) Heatmap of the top 100 most dysregulated genes (50 up- and 50 down-regulated). Columns represent samples (tumor vs. normal); rows represent genes clustered by expression. Expression Z-scores are shown from blue (low) to red (high). (C) Circular enrichment plot of DEGs. From outer to inner: term IDs; number of background genes per term; number of DEGs enriched per term; rich factor. (D, F) Top 15 significant GO terms (p < 0.05) for (D) up- and (F) down-regulated DEGs. Bar length indicates gene count; color indicates ontology. (E, G) Significant KEGG pathways (p < 0.05) for (E) up- and (G) down-regulated DEGs. Dot size indicates number of enriched genes; color indicated -log10 (adjusted p-value). All analyses were performed using the Home for Researchers platform (https://www.home-for-researchers.com/) based on TCGA-HNSC data.
(TIF)

**S2 Fig. Protein-protein interaction (PPI) network of paraben-related targets in HNSCC.** The PPI network was constructed from 80 paraben-related targets using STRING, resulting in 568 interactions. Nodes represent proteins; edges represent functional associations. Edge color indicates different known or predicted interactions. The network shows significant functional connectivity (PPI enrichment p < 1.0 × 10–16).
(TIF)

**S3 Fig. Protein expression and subcellular localization of hub genes in HNSCC.** Immunohistochemistry images and subcellular localization data for the hub genes were retrieved from the Human Protein Atlas (HPA) database. For each panel (A: CCNA1, B: CDK1, C: CCNA2, D: CDK2, E: CDK4, F: TYMS, G: AURKA, H: CCNA1, I: CCNB2, J: PLK1.), the left image shows target protein expression in normal tissue (oral mucosa), while the middle image shows expression in HNSC tissue (not detected, low, medium, or high). Subcellular localization of the corresponding core proteins in U2OS cells is displayed right each pair (the blue represents the target protein). In this analysis, CDC25A and CHEK1 expression

were not included in HPA, subcellular localization of TYMS and PLK1 were also not included in it. All the detailed data can be acquired from HPA database.
(TIF)

**S4 Fig. Proteomic analysis of hub genes in HNSC via UALCAN-CPTAC.** Bar plots compare normalized protein expression levels between normal (n = 71) and HNSCC tumor (n = 108) samples from CPTAC data. Wilcoxon test was used; p < 0.05 was considered significant. CCNA1 and CDC25A were not available in the dataset.
(TIF)

**S5 Fig. mRNA expression levels of hub genes in HNSCC via GEPIA3.** The mRNA expression of the 12 hub genes was analyzed in HNSCC using the GEPIA3 database. Expression levels are displayed as log2(TPM + 1). Red box plots represent tumor tissues (n = 520), and green box plots represent normal tissues (n = 44). All 12 hub genes showed significantly elevated expression in tumor tissues compared to normal controls (p < 0.001, unpaired Student's t-test).
(TIF)

**S6 Fig. Survival analysis of hub genes in HNSCC.** Overall survival analysis of 12 hub genes was performed using the GEPIA3 database based on TCGA-HNSC data. Gene expression was quantified as log2(TPM + 1). Cox proportional hazards regression analysis revealed that among all hub genes, only high expression of CCNA1 was significantly associated with poor overall survival (p < 0.05). Survival time is shown in months. The Kaplan-Meier curve demonstrates the stratification of patients based on CCNA1 expression levels (high vs. low). TPM: transcripts per million.
(TIF)

**S7 Fig. Correlation between hub gene expression and immune cell infiltration in HNSCC.** Analysis was performed using the TIMER2.0 database to evaluate the association between expression levels of 12 hub genes and the abundance of immune cell infiltration (CD8 + T cells, CD4 + T cells, and B cells) in HNSCC. A heatmap with numbers displays the purity-adjusted Spearman's correlation coefficients (ρ-values) across multiple algorithms in HNSCC. Red indicates statistically significant positive correlation (ρ > 0, p < 0.05), light blue indicates statistically significant negative correlation (ρ < 0, p < 0.05), and light gray indicates no significant correlation (p > 0.05). Multiple algorithms consistently showed that CCNA1 expression was negatively correlated with CD8 + T cell and B cell infiltration levels.
(TIF)

**S1 Table. Toxicity prediction results of the selected parabens.**
(DOCX)

**S2 Table. The results for Venn-diagram.**
(XLSX)

**S3 Table. The top 10 terms of each category in GO and KEGG analyses.**
(XLSX)

**S4 Table. Enrichment analyses of hub genes.**
(XLSX)

## Author contributions

**Conceptualization:** Baoshan Wang, Lei Zhao.

**Data curation:** Tao Liu, Huan Cao, Miaomiao Yu.

**Formal analysis:** Lei Zhao, Tao Liu.

**Funding acquisition:** Baoshan Wang.

**Investigation:** Huan Cao.

**Methodology:** Lei Zhao, Jianwang Yang.

**Project administration:** Baoshan Wang, Lei Zhao.

**Resources:** Lei Zhao, Jianwang Yang.

**Software:** Lei Zhao, Jianwang Yang, Tao Liu.

**Supervision:** Baoshan Wang, Lei Zhao.

**Visualization:** Lei Zhao, Miaomiao Yu.

**Writing – original draft:** Lei Zhao.

**Writing – review & editing:** Lei Zhao, Jianwang Yang, Tao Liu, Huan Cao, Miaomiao Yu.

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
