## [Decision Letter · Decision Letter 0]

26 Feb 2026

Dear Dr. Wang,

We look forward to receiving your revised manuscript.

Kind regards,

Amel Mohamed El Asely

Academic Editor

PLOS One

Journal Requirements:

This research was funded by the Clinical Research Center Foundation of Hebei Provincial Department of Science and Technology, grant number 20577716D, and the Natural Science Foundation of Hebei Province, grant number H2022206376.

Reviewers' comments:

Reviewer's Responses to Questions

**Comments to the Author**

1. Is the manuscript technically sound, and do the data support the conclusions?

Reviewer #1: Yes

2. Has the statistical analysis been performed appropriately and rigorously?

Reviewer #1: Yes

3. Have the authors made all data underlying the findings in their manuscript fully available?

Reviewer #1: Yes

4. Is the manuscript presented in an intelligible fashion and written in standard English?

Reviewer #1: Yes

Reviewer #1: Naive idea, good research methodology

, scientific writing is excellent, very creative model of investigation

The selected sample size are SCC head and neck pathology, can be specified to one anatomical site like larynx for example.

**Do you want your identity to be public for this peer review?** For information about this choice, including consent withdrawal, please see our Privacy Policy

Reviewer #1: **Yes:** Nehal mohamed Elmashad

---

## [Author Response · Author response to Decision Letter 1]

4 Mar 2026

Dear Academic Editor Amel Mohamed El Asely and Reviewers

We would like to thank you and the reviewers for your thoughtful and constructive comments on our manuscript (PONE-D-25-51064, Title: Elucidating the potential carcinogenic molecular mechanisms of parabens in head and neck squamous cell carcinoma through network toxicology and molecular docking). We have carefully considered all the suggestions and have revised the manuscript accordingly. Below is a point-by-point response to the reviewers' comments. All changes in the manuscript are marked using track changes for your convenience.

For Journal Requirements:

Response: We have carefully reviewed the PLOS ONE formatting templates (both the main body sample and the title/author sample). We have thoroughly revised our manuscript to ensure full compliance with all of the journal's style requirements. The key modifications we have made include: Manuscript body formatting, Figure and table citations, Reference list, Title page and Author affiliations, File naming, and so on. All changes made to meet these requirements have been clearly marked using the "Track Changes" feature in the revised manuscript file. We believe the manuscript now fully adheres to PLOS ONE's formatting policies and thank the editor for the opportunity to correct these details.

2. Please note that PLOS One has specific guidelines on code sharing for submissions in which author-generated code underpins the findings in the manuscript.

Response: We have reviewed the code-sharing guidelines. The computational analysis in this study was performed using publicly available software and online databases (e.g., CTD, GeneCards, SwissTargetPrediction, AutoDock Tools). We don't have any special scripts to share.

3. Thank you for stating the following financial disclosure: This research was funded by the Clinical Research Center Foundation of Hebei Provincial Department of Science and Technology, grant number 20577716D, and the Natural Science Foundation of Hebei Province, grant number H2022206376. Please state what role the funders took in the study.

Response: Thank you for this reminder. We have included the following statement in our cover letter. Please update our financial disclosure accordingly: "The funders had no role in study design, data collection and analysis, decision to publish, or preparation of the manuscript."

Response: We have reviewed the reviewer’s comments. No specific publications were recommended for citation. If the editor or reviewer has any specific references in mind that they feel are essential, we would be happy to review and incorporate them in a further revision.

5. Please review your reference list to ensure that it is complete and correct. If you have cited papers that have been retracted, please include the rationale for doing so in the manuscript text, or remove these references and replace them with relevant current references.

Response: We have carefully reviewed our entire reference list. We confirm that all references are complete, accurate, and relevant to the current study. We have verified that no retracted articles are cited in this manuscript. Any minor formatting errors in the reference list have been corrected according to the PLOS ONE style.

For Reviewers' comments:

1. Is the manuscript technically sound, and do the data support the conclusions?

Reviewer #1: Yes

Response: We sincerely thank the reviewer for this positive assessment of our technical approach and the validity of our conclusions. We have maintained this technical rigor throughout our revisions.

2. Has the statistical analysis been performed appropriately and rigorously?

Reviewer #1: Yes

Response: We appreciate the reviewer's confirmation of our statistical methodology. We have ensured that all statistical analyses remain appropriate and rigorous in the revised version.

3. Have the authors made all data underlying the findings in their manuscript fully available?

Reviewer #1: Yes

Response: Thank you for this confirmation. We have further strengthened our data availability statement to explicitly indicate that all relevant data are within the paper and its Supporting Information files.

4. Is the manuscript presented in an intelligible fashion and written in standard English?

Reviewer #1: Yes

Response: We are grateful for the reviewer's positive comment on our scientific writing. We have carefully reviewed the entire manuscript for any typographical or grammatical errors and have made minor corrections to ensure the language remains clear, correct, and unambiguous.

5. Review Comments to the Author

Reviewer #1: Naive idea, good research methodology, scientific writing is excellent, very creative model of investigation. The selected sample size are SCC head and neck pathology, can be specified to one anatomical site like larynx for example.

Response: We sincerely thank the reviewer for this insightful suggestion. We fully agree that HNSCC is heterogeneous and that focusing on a specific subsite like the larynx would enhance clinical precision. However, we encountered a technical limitation: the public toxicogenomic databases (CTD, GeneCards, TCGA Pan-Cancer) used in our pipeline are designed for broad screening and do not currently allow precise subsite filtering without introducing significant selection bias or compromising network integrity.

To ensure scientific transparency while respecting the reviewer's valuable input, we have strengthened the manuscript in "Discussion" section (Page 20, Lines 419-425). In this section, we explicitly acknowledge that the use of aggregated HNSCC data may mask subsite-specific molecular mechanisms. This plan specifically calls for the use of TCGA subsite-specific cohorts (laryngeal cancer data) to validate and refine the targets identified in our current study. We also suggest experimental validation in subsite-relevant cell lines.

We are grateful to the reviewer for guiding us to address this important issue, which has ultimately made our manuscript more nuanced and academically rigorous.

6. PLOS authors have the option to publish the peer review history of their article (what does this mean?). If published, this will include your full peer review and any attached files.

Reviewer #1: Yes: Nehal mohamed Elmashad

Response: We acknowledge the reviewer's consent to publish their identity with the peer review history. We have no objection to this.

7. To ensure your figures meet our technical requirements, please review our figure guidelines. You may also use PLOS’s free figure tool, NAAS, to help you prepare publication quality figures.

Response: Thank you for directing us to the NAAS figure preparation tool and the detailed figure guidelines. We have taken the following steps to ensure our figures meet PLOS ONE's publication quality standards:

NAAS Tool Assessment: We have uploaded all our figure files to the PLOS NAAS tool. Based on the feedback provided by the tool, we have made necessary adjustments to resolution, color mode, font sizes, and file formats.

Figure File Naming and Format: All figure files have been renamed according to the guidelines (e.g., "Fig1.tif", "Fig2.tif") and saved in the recommended TIFF format with appropriate compression and resolution (at least 300 dpi).

Figure Captions and Citations: We have double-checked that all figure citations in the text (e.g., "Fig 1", "Figs 2 and 3") appear in the correct order and that each figure caption is complete, includes a title and legend, and is placed directly after the paragraph of first citation, as per the manuscript body formatting guidelines.

We confirm that all figures now comply with PLOS ONE's technical specifications.

We hope that these revisions fully address all the points raised by the reviewer and the journal. We are confident that the changes have substantially improved the manuscript and we look forward to hearing from you regarding its acceptance.

Sincerely,

Baoshan Wang

---

## [Editor Report · Decision Letter 1]

8 Mar 2026

Elucidating the potential carcinogenic molecular mechanisms of parabens in head and neck squamous cell carcinoma through network toxicology and molecular docking

PONE-D-25-51064R1

Dear Dr. Baoshan Wang,

We’re pleased to inform you that your manuscript has been judged scientifically suitable for publication and will be formally accepted for publication once it meets all outstanding technical requirements.

Kind regards,

Amel Mohamed El Asely

Academic Editor

PLOS One
---

## [Editor Report · Acceptance letter]

PONE-D-25-51064R1

PLOS One

Dear Dr. Wang,

I'm pleased to inform you that your manuscript has been deemed suitable for publication in PLOS One. Congratulations! Your manuscript is now being handed over to our production team.

Kind regards,

on behalf of

Prof. Amel Mohamed El Asely

Academic Editor

PLOS One